# The Inhibitory Effect of Propylene Glycol Alginate Sodium Sulfate on Fibroblast Growth Factor 2-Mediated Angiogenesis and Invasion in Murine Melanoma B16-F10 Cells In Vitro

**DOI:** 10.3390/md17050257

**Published:** 2019-04-29

**Authors:** He Ma, Peiju Qiu, Huixin Xu, Ximing Xu, Meng Xin, Yanyan Chu, Huashi Guan, Chunxia Li, Jinbo Yang

**Affiliations:** 1Key Laboratory of Marine Drugs of Ministry of Education, Shandong Provincial, Key Laboratory of Glycoscience and Glycotechnology, School of Medicine and Pharmacy, Ocean University of China, Qingdao 266003, China; mahe1992mahe@163.com (H.M.); xvhuixin@hotmail.com (H.X.); xuximing@ouc.edu.cn (X.X.); xinmeng512@126.com (M.X.); mary0312332@126.com (Y.C.); hsguan@ouc.edu.cn (H.G.); lchunxia@ouc.edu.cn (C.L.); 2Innovation Center for Marine Drug Screening & Evaluation, Pilot National Laboratory for Marine Science and Technology (Qingdao), Qingdao 266237, China; 3Laboratory for Marine Drugs and Bioproducts of Pilot National Laboratory for Marine Science and Technology (Qingdao), Qingdao 266237, China; 4Marine Biomedical Research Institute of Qingdao, Qingdao 266071, China

**Keywords:** propylene glycol alginate sodium sulfate, angiogenesis, invasion, FGF2, MMP-2, MMP-9

## Abstract

Melanoma is one of the most malignant and aggressive types of cancer worldwide. Fibroblast growth factor 2 (FGF2) is one of the critical regulators of melanoma angiogenesis and metastasis; thus, it might be an effective anti-cancer strategy to explore FGF2-targeting drug candidates from existing drugs. In this study, we evaluate the effect of the marine drug propylene glycol alginate sodium sulfate (PSS) on FGF2-mediated angiogenesis and invasion. The data shows that FGF2 selectively bound to PSS with high affinity. PSS inhibited FGF2-mediated angiogenesis in a rat aortic ring model and suppressed FGF2-mediated invasion, but not the migration of murine melanoma B16-F10 cells. The further mechanism study indicates that PSS decreased the expression of activated matrix metalloproteinase 2 (MMP-2) and matrix metalloproteinase 9 (MMP-9), and also suppressed their activity. In addition, PSS was found to decrease the level of Vimentin in B16-F10 cells, which is known to participate in the epithelial–mesenchymal transition. Notably, PSS did not elicit any changes in cancer cell viability. Based on the results above, we conclude that PSS might be a potential drug to regulate the tumor microenvironment in order to facilitate the recovery of melanoma patients.

## 1. Introduction

Melanoma is one of the most malignant and aggressive types of cancer worldwide, and the identification of new targets for treating melanoma is urgently needed. Melanoma cells have been reported to constitutively express fibroblast growth factor 2 (FGF2) [1,2], which is an autocrine factor and promotes the proliferation, angiogenesis, and metastasis of melanoma cells. Thus, FGF2 is commonly considered one of the potential targets for treating melanoma, and may be used with other targets to synergistically enhance therapeutic efficacy [3].

FGF2, which belongs to the FGF family, participates in a variety of physiological and pathological processes both in vitro and in vivo, including cellular survival, differentiation, proliferation, angiogenesis, adhesion, skeletal formation, and wound healing [4,5,6]. In terms of function, FGF2 first binds to two fibroblast growth factor receptors (FGFRs), and then recruits the heparan sulfate (HS) chain(s) of membrane-anchored heparan sulfate proteoglycan (HSPGs) for assembly into a ternary complex (FGF–HS–FGFR). The FGF–HS–FGFR complex results in receptor dimerization, with subsequent autophosphorylation of specific tyrosine residues that affects multiple downstream signal transduction pathways [7,8,9,10]. Meanwhile, FGF2 and FGFR-1 complexes can enter into the nucleus, where they engage with a variety of sequence-specific transcription factors and further regulate the release and activity of matrix metalloproteinase 2 (MMP-2) and matrix metalloproteinase 9 (MMP-9), as well as the expression of proteins involved in the epithelial–mesenchymal transition [11,12].

Among the FGF–HS–FGFR complex, the basic amino acid residues of FGF2–FGFR2 form a positively charged cluster, which can attract negatively-charged HS chains for incorporation. Exogenous sulfated carbohydrates like heparin and its mimetic derivatives have long been studied for their competition with HS chains, to break the formation of the FGF2–HS–FGFR1 ternary complex and physiologically disrupt the function of cancer cells [13]. Several heparin mimics have been proven to be effective in blocking the formation of the FGF2–HS–FGFR1 ternary complex in vitro; however, their potential use might be limited by their potency, pharmacokinetic defect, and safety profile. Therefore, it would be an effective strategy to explore FGF2 inhibitors from existing drugs to facilitate cancer treatment.

Propylene glycol alginate sodium sulfate (PSS) is a heparin-like drug that was approved by the China Food and Drug Administration (CFDA) over 30 years ago to treat hyperlipidemia and ischemic cardio-cerebrovascular diseases [14]. PSS is obtained from alginate polysaccharide of Laminaria with multiple chemical modifications. PSS is composed of mannuronic acid (M) and guluronic acid (G) disaccharide repeat units and sulfates occurring at the C-2 or C-3 position of the sugar moiety, with a substitution at the C-6 position by propylene glycol. PSS has the following structural characteristics: an M/G ratio above 1.5, a molecular weight of 15–20 kDa, and an organic sulfur content of 9–14% [15].

Based on our previous research, it is well known that PSS and its fractions exert effects on anti-coagulation-related activities and anti-selectin activities [15,16,17]. Wu et al. reported that PSS and its oligosaccharides could significantly stimulate FGF2-induced cell proliferation in FGFR1c-expressing BaF3 murine pro-B cell line [18], suggesting that the potential binding effect occurs between PSS and FGF2. However, no studies have investigated the effects of PSS on FGF2-mediated functional regulation on a highly metastatic B16-F10 melanoma model and the related tumor microenvironment. As PSS possesses similar structural and bioactive properties to heparin, we hypothesized that PSS might exhibit an inhibitory effect on FGF2-mediated cellular proliferation, invasion, migration, or angiogenesis, as well as related downstream signaling.

## 2. Results

### 2.1. Fibroblast Growth Factor 2 Bound to Propylene Glycol Alginate Sodium Sulfate with High Affinity

PSS was investigated by surface plasmon resonance (SPR) analysis for its affinity with FGF2 and vascular endothelial growth factor 165 (VEGF165). As shown in Figure 1, PSS bound directly to FGF2, and the equilibrium dissociation constant (KD) was 2.73 × 10^−8^ M, which is comparable to heparin (2.76 × 10^−8^ M). In contrast, weak affinity was detected between PSS and VEGF165 (Appendix A), while the KD of heparin was 8.09 × 10^−7^ M. As FGF2 is a crucial growth factor to regulate angiogenesis and the function of tumor cells, these data indicate that FGF2 was probably a potential target for PSS to improve tumor environment.

### 2.2. Propylene Glycol Alginate Sodium Sulfate Inhibited the Fibroblast Growth Factor 2-Induced Invasion of B16-F10 Cells

FGF2 has been reported to play an important role in tumor metastasis; thus, we next wanted to evaluate the role of PSS in inhibiting tumor cell invasion and metastasis. PSS significantly inhibited the serum-induced invasion of B16-F10 in a dose-dependent manner. The Matrigel barrier approach was used to evaluate tumor cell metastasis. As shown in Figure 2A,B, after treating the cells for 16 h, PSS inhibited invasion by 33.8%, 45.7%, and 61.8% at concentrations of 25, 50, and 100 μg/mL, respectively, in a dose-dependent manner. We further detected the anti-invasive effect of PSS on FGF2-mediated invasion. As shown in Figure 2C,D, 10% FBS and 200 ng/mL FGF2 induced a comparable number of B16-F10 cells to penetrate the growth factor-reduced Matrigel, and PSS significantly inhibited FGF2-induced invasion by 59.1% at a concentration of 50 μg/mL after treating the cells for 8 h.

### 2.3. Propylene Glycol Alginate Sodium Sulfate Had No Effect on Cell Viability

To ensure that the inhibitory effect of PSS on invasion was not due to the direct killing of tumor cells by PSS, we detected the proliferation ability of tumor cells after treatment with PSS for 48 h. No inhibition on proliferation was observed, even when the concentration was more than 1000 μg/mL (Figure 3), suggesting that the inhibitory effect of PSS on invasion was not due to a reduction in the viability of cancer cells, but was likely due to the inhibition of tumor cell migration or the suppression of matrix-degrading enzymes.

### 2.4. Propylene Glycol Alginate Sodium Sulfate Had No Effect on the Migration of B16-F10 Cells

Based on the results above, we further detected the effect of PSS on the migration of B16-F10 cells. As shown in Figure 4A,B, no obvious difference in the number of the migratory cells was found between the untreated control and the treatment with PSS at 400 μg/mL. Meanwhile, the results of the scratch wound migration assays also revealed no significant difference across the wounded region after treatment with PSS at 400 μg/mL for 24 h (Figure 4C). PSS also had no effect on the migration of human umbilical vein endothelial (HUVEC) cells (Appendix A).

### 2.5. Propylene Glycol Alginate Sodium Sulfate Down-Regulated the Expression of Activated Matrix Metalloproteinase 2 and Matrix Metalloproteinase 9

The process of invasion involves the degradation of the ECM and the subsequent migration of tumor cells. Based on the data above, we confirmed that PSS inhibited the invasion of B16-F10 cells in a dose-dependent manner, but had no effect on tumor cell migration. Therefore, we examined whether PSS affected the expression or activity of crucial ECM-degrading enzymes. MMP-2 and MMP-9 are capable of degrading type IV collagen, which is the most abundant component of the basement membrane. Degradation of the basement membrane is an essential step for the metastatic progression of most cancers. Matrigel is an analog of the basement membrane. PSS inhibits the degradation of Matrigel by B16-F10. Therefore, it is necessary to detect whether PSS inhibits the expression and activity of MMP-2 and MMP-9. We first detected the effect of PSS on the expression of MMP-2 and MMP-9. After treating B16-F10 cells for 24 h, PSS was found to decrease the expression of activated MMP-2 in a dose-dependent manner. PSS inhibited the level of MMP-2 by 1%, 15%, 35%, and 67% compared to the untreated control at concentrations of 12.5, 25, 50, and 100 µg/mL (Figure 5A,B), respectively. PSS also exerted an inhibitory effect on the expression of MMP-9. As shown in Figure 5C,D, PSS at a concentration of 100 µg/mL suppressed the expression of MMP-9 by 32%.

### 2.6. Propylene Glycol Alginate Sodium Sulfate Decreased the Activity of Matrix Metalloproteinase 2 and Matrix Metalloproteinase 9

We further detected the activity of MMP-2 and MMP-9 in B16-F10 cells after treatment with PSS for 24 h. As shown in Figure 6B,C, PSS inhibited the activity of MMP-2 in a dose-dependent manner. At 25 μg/mL, PSS slightly inhibited MMP-2, while at 50 and 100 μg/mL, it suppressed the activity by 21.8% and 28.7%, respectively. PSS also suppressed the activity of MMP-9 in a similar manner. The inhibition of PSS was 2.7%, 17.2%, and 25.1% at concentrations of 25, 50, and 100 μg/mL, respectively (Figure 6E,F).

### 2.7. Propylene Glycol Alginate Sodium Sulfate Down-Regulated the Protein Expression of Vimentin

Carcinoma cells make use of the epithelial–mesenchymal transition (EMT) as they become invasive. Indeed, the transition typically features the loss of cell–cell adherence proteins like cadherin, followed by the loss of apico-basal polarity, and finally, gaining the ability to migrate and invade. To investigate the potential role of PSS in the EMT, two crucial EMT markers in B16-F10 cells were examined by western blot. The expression of Vimentin was decreased by 41% and 47% in the presence of PSS at concentrations of 50 and 100 μg/mL, respectively (Figure 7), while a slight increase was detected in the level of E-cadherin. Additionally, other proteins involved in the regulation of invasion and migration did not show any change in expression after treatment with PSS.

### 2.8. Propylene Glycol Alginate Sodium Sulfate Inhibited Angiogenesis

We next assessed the capacity of PSS to inhibit angiogenic activity using rat aortic rings and chick chorioallantoic membrane models. As shown in Figure 8A,B, PSS inhibited the outgrowth of new microvessels in a dose-dependent manner. The molecular weight of sulfated polysaccharides including PSS played crucial roles to determine their bioactivities. To illuminate the interaction between molecular weight and anti-angiogenesis potency of PSS, we further detected the effects of PSS fractions with various molecular weights (Mw) (H1: Mw 21.91 kDa; H3: Mw 13.68 kDa; H5: Mw 6.56 kDa; H7: Mw 3.10 kDa; H8: Mw 2.26 kDa) [17] on new blood vessel formation, using the same model. The data showed that the inhibitory effect of the fractions closely correlated with the molecular weight. The H1 fraction (Mw = 21.91 kDa) exhibited the most potent effect among all five fractions, and the inhibitory effect was reduced as the molecular weight decreased (Figure 8C,D). Meanwhile, in the chick chorioallantoic membrane model, PSS suppressed vessel formation at doses of 100 and 200 μg/egg (Figure 8E,F).

### 2.9. Propylene Glycol Alginate Sodium Sulfate Inhibited Fibroblast Growth Factor 2-Mediated Angiogenesis

To confirm the possible effect of PSS on FGF2-mediated angiogenesis, we cultured rat aortic rings in a serum-free medium, with 200 ng/mL FGF2 in the plate coating with growth factor-reduced Matrigel. As shown in Figure 9, little sprouting was observed in the negative control ring, while substantially more sprouting was observed in the FGF2-treated ring. PSS at 100 μg/mL decreased the level of sprouting to the amount in the negative control, and 200 μg/mL completely suppressed sprouting.

## 3. Discussion

In this study, we observed that PSS has a major impact on invasion and angiogenesis in murine melanoma B16-F10 cells. FGF2 is a proangiogenic factor involved in tumor angiogenesis, invasion and migration. The present findings show that FGF2 bound to PSS with high affinity and inhibited FGF2-mediated angiogenesis in a rat aortic ring model. Moreover, PSS could suppress a FGF2-mediated invasion. The results of a further mechanism study indicate that PSS down-regulated the expression of activated MMP-2 and MMP-9, and also suppressed their activity. In addition, PSS was found to decrease the protein levels of Vimentin, which is known to participate in EMT. Notably, PSS did not elicit any changes in cancer cell viability, even though the concentration was more than 1000 μg/mL.

Previous data indicated that the KD value for heparin binding to FGF2 ranges from 1 to 71 nM [19,20,21,22]. These values were affected by a variety of factors, including the method used to determine them, the ionic strength of the buffer, the size and source of the heparin, and the source of the growth factor. In our system, we obtained similar KD values for FGF2 binding to PSS or heparin, indicating that PSS was comparable to heparin in terms of binding to FGF2. We further analyzed the electrostatic potential surface of FGF2. As shown in Appendix A, FGF2 displayed a mass area of positive charge in the surface; hence, it could easily bind negative charged compounds. PSS and heparin possess similar sulfate group contents (32.39% and 34%, respectively) and similar KD values to FGF2; therefore, we presume that the sulfate of the two polysaccharides probably accounted for their affinity to FGF2.

FGF2 is known to interact with *N*-sulfoglucosamine (GlcNS) and 2-*O*-sulfated iduronate residues (IdoUA (2S)) in heparin and HS [23,24], but the additional presence of 6-*O*-sulfation is required for biological activity [25,26]. PSS is a heparin-like drug, which is composed of repeating units of mannuronic acid (M) and guluronic acid (G), with 2-*O* and 3-*O* sulfate groups in the sugar rings. Groups that are 2-*O*-sulfated play a crucial role in mediating the binding of heparin with FGF2. Because PSS and heparin exhibited a comparable affinity to FGF2 and, we presume that 2-*O*-sulfated PSS might also be crucial for promoting the interaction of PSS with FGF2. To confirm this presumption, further research should be performed to elucidate the structure–activity relationship.

FGF2 and VEGF165 are the most important growth factors, and can be blocked by heparin to reduce angiogenesis. We also detected the affinity of PSS with VEGF165; however, PSS exhibited weaker affinity to VEGF165 (KD = 1.78 × 10^−4^ M) than heparin (KD = 8.09 × 10^−7^ M). Zhao et al. [27] reported that the specific structural features of heparin, such as the content of sulfate, sugar ring stereochemistry, and conformation, determined the affinity of heparin-derived oligosaccharides to VEGF165. Moreover, the positive charge on the surface of VEGF165 was distributed in a dispersed state (Appendix A). Based on the above information, we presume that the conformation of PSS might not fit the stereochemical structure of VEGF165.

It was well-documented that the molecular weight of sulfated polysaccharides including PSS played crucial roles in determining their bioactivities. Our previous study showed that the average molecular weight of PSS was about 17 kDa and the distribution range of molecular weight was about 2~20 kDa [17]. Unlike the small molecular compounds, which generally bind to domains with catalytic activity of targeting proteins, sulfated polysaccharides possessed large amount of negative charge and generally interacted with proteins rich in positive potential on the surface of proteins. Theoretically, the longer sugar chains (the higher molecular weight) of the negative charged polysaccharide was endowed with the stronger binding affinity to target proteins and further exerted obvious bioactivities. Here, the higher molecular weight of PSS fractions exerted stronger inhibitory effect on angiogenesis which was consistent with the trend in previous publications [17,28]. Similarly, we speculated that it might be the same trends of PSS fractions in other experiments of the manuscript.

For the first time, we evaluated the effect of PSS on the highly metastatic B16-F10 melanoma cells and the related tumor environment. PSS itself has no inhibitory effect on the growth of B16-F10 cells—however, it suppressed FGF2-mediated angiogenesis and invasion of B16-F10 cells, and also decreased the level of Vimentin, which might help enhance the sensitivity of tumor cells to chemotherapy. Moreover, to fully elucidate the effects of PSS on the tumor microenvironment, further research should be conducted to investigate whether PSS exerts inhibitory effects on other cells involved in the tumor microenvironment, such as endothelial cells, fibroblasts, and immune cells. Meanwhile, further research should be done to combine PSS with chemotherapeutic drugs to check whether a synergistical effect happens.

## 4. Materials and Methods

### 4.1. Cell Culture and Reagents

The murine melanoma B16-F10 cell line was all obtained from the Type Culture Collection of the Chinese Academy of Sciences (Shanghai, China). The B16-F10 cells were cultured in RPMI-1640 supplemented with 10% (*v*/*v*) heat-inactivated FBS and 1% (*v*/*v*) penicillin–streptomycin. HUVECs were obtained from the Procells company (Shanghai, China) and cultured in Ham’s F-12K supplemented with 100 μg/mL Heparin, 50 μg/mL ECGs, 10% (*v*/*v*) heat-inactivated FBS, and 1% (*v*/*v*) penicillin-streptomycin solution. Cells were cultured at 37 °C in a humidified atmosphere containing 5% CO_2_. Cells were maintained at subconfluency, and the culture medium was changed every other day. The B16-F10 cells used were between 3 and 30 passages.

PSS was provided by Chia Tai Haier Pharmaceutical Co., Ltd. (Qingdao, China). Heparin (201 U/mg) was obtained from Wanbang Pharmaceuticals Company (Xuzhou, China). All proteins were purchased from Sino Biological (Beijing, China). All antibodies were purchased from Cell Signaling Technology (Cell Signaling Technology Inc., MA, USA). Matrigel was purchased from Corning Company (Tewksbury, MA, USA). All other chemicals and solvents were of analytical grade and purchased from Sinopharm Group Co. Ltd. (Beijing, China)

### 4.2. The Binding Kinetics of Propylene Glycol Alginate Sodium Sulfate and Fibroblast Growth Factor 2

The kinetics and specificity of the binding between PSS derivatives and FGF2 and VEGF165 proteins were determined by a PlexArray^®^HT SPR system (Plexera Inc., Seattle, DC, USA). Briefly, PSS (1–5 mM) were immobilized on Graft-to-PCL sensor chips by UV crosslinking for 15 min, according to an established protocol. The mobile phase was FGF2 or VEGF165 solution (dissolved in PBS), and the concentrations used were 125, 250, and 500 nM. The data obtained were analyzed and fitted by PLEXERA SPR DAM to obtain the equilibrium dissociation constant (KD).

### 4.3. Cell Invasion

Transwell chambers (6.5 mm diameter, 8 μm pore size; Corning Life Sciences) were coated with 100 μL of diluted Matrigel. Then, 0.6 mL of medium containing 10% FBS was added to the lower chambers, and cells suspended in serum-free medium at a density of 1.5 × 10^5^ cells/mL were seeded (0.1 mL) in the upper chambers. Various concentrations of PSS (25, 50, 100 μg/mL) were added to both of the upper and lower chambers. After incubation for 16 h or 8 h (FGF2-mediated invasion of B16-F10), cells were fixed with cold 4% paraformaldehyde and stained with 0.1% crystal violet, and the cells that had not migrated were removed from the upper chambers. The remaining cells were photographed in five random fields per membrane. The dye was dissolved in 80 μL of acetic acid, and the absorbance of the resulting solution was measured at 600 nm using a microplate reader (SpetraMAX i3, Molecular Devices, Sunnyvale, CA, USA).

### 4.4. Cell Proliferation Assay

B16-F10 cells (0.5 × 10^4^ cells/well) were seeded in 96-well culture plates in 100 μL of culture medium and incubated for 24 h. Subsequently, 100 μL of complete medium without or with various concentrations of PSS (100, 200, 400, 600, 800, 1000 μg/mL) were added. After incubation for 48 h, 10 μL of resazurin solution (1 mg/mL) was then added to each well, and the cells were incubated for another 4 h. The fluorescence of each well was measured at 544 nm and 595 nm by a microplate reader (SpetraMAX i3, Molecular Devices, Sunnyvale, CA, USA).

### 4.5. Cell Migration

First, 0.6 mL medium containing 10% FBS was added to the lower chamber of Transwell chambers (6.5 mm diameter, 8 μm pore size; Corning Life Sciences), and cells suspended in a serum-free medium at a density of 1.5 × 10^5^ cells/mL were seeded (0.1 mL) in the upper chambers. Then, 400 μg/mL PSS was added to both the upper and lower chambers. After incubation for 16 h, the cells were fixed by cold 4% paraformaldehyde, stained by 0.1% crystal violet, and cells that had not migrated were removed from the upper chambers. The remaining cells were photographed in five random fields per membrane. The dye was dissolved in 80 μL of acetic acid, and the absorbance of the resulting solution was measured at 600 nm using a microplate reader (SpetraMAX i3, Molecular Devices, Sunnyvale, CA, USA).

### 4.6. The Wound Healing Assay

The effect of PSS on migration was analyzed in vitro using a wound healing assay. B16-F10 cells were seeded in 12-well culture plates to reach 70% confluency. The cell monolayer was scratched vertically down the center of each well with a sterile 200 μL micropipette tip, and rinsed carefully with phosphate buffer solution (PBS) three times to remove cell debris. FBS-free medium with varying concentrations of PSS (100, 400 μg/mL) or 400 μg/mL heparin was added to each well. Three randomly selected views along the wound line in each well were photographed under an inverted microscope at 0 h and 24 h after incubation. The percentage of void area with respect to time 0 was determined using ImageJ software (ImageJ 1.8.0, Rawak Software Inc., Stuttgart, Germany).

### 4.7. Western Blot Analysis

B16-F10 cells (2 × 10^6^ cells per well) were seeded into a 10 cm dish for 24 h, and then cells were treated with different concentrations of PSS (12.5, 25, 50, 100 μg/mL) for 24 h. The medium was removed, and the cells were washed with PBS three times. Cells were then lysed in 200 μL of lysis buffer on ice. The total protein was determined using the Bicinchoninic Acid (BCA) Kit (Solarbio, Beijing, China). Equal amounts of protein in the cell extracts were fractionated by 10% SDS-PAGE, and then electrotransferred onto polyvinylidene fluoride (PVDF) membranes. After blocking with TBST (20 mM Tris-buffered saline and 0.1% Tween) containing 5% nonfat dry milk for 1 h at room temperature, the membranes were incubated for 2 h with monoclonal antibodies, such as anti-MMP-9, anti-MMP-2, anti-E-cadherin, anti-Vimentin, anti-ERK 1/2, anti-p-ERK 1/2, anti-AKT, anti-p-AKT (Ser473), anti-p38, anti-p-p38, anti-p-p38, anti-NF-κB, anti-p-NF-κB, and anti-β-actin, which were purchased from Cell Signaling Technology. The membranes were then washed three times and incubated with HRP-conjugated secondary antibodies (Abcam, Cambridge, Massachusetts, USA). The proteins were then detected using chemiluminescence agents (Amersham ECL, GE Healthcare, Buckinghamshire, UK).

### 4.8. In Situ Zymography Localization of Matrix Metalloproteinase 2 and Matrix Metalloproteinase 2 Activity

MMP-2/9 activity was tested on the slides of cells using the GENMED Kit (Genmed Scientifics Inc., Wilmington, DE, USA) according to the manufacturer’s instructions (GENMED80062.4/80062.6, GENMED). B16-F10 were seeded in 12-well culture plates with a glass slide until they reached 50% confluence. After washing with PBS, fresh serum-free culture media was added to the plate in the presence or absence of serial PSS treatments, at concentrations ranging from 25–100 μg/mL. After 24 h, the glass slides were carefully removed. Reagent A was heated until melted. Then, 1000 μL of reagent A was transferred into a 1.5 mL microtube and incubated for 10 min at 37 °C in a thermostatic water bath. Reagent B was added to the 1.5 mL microtube and mixed well. The solution was added to the slide, a coverslip was added, and the slide was incubated in the dark at 4 °C for 10 min until the gel solidified. The prepared sections were then incubated at 37 °C for 60 min in the dark. Fluorescence was visualized in 10 randomly selected fields of view for each cell slide at 40× magnification under a fluorescence microscope (Colibri 7, ZEISS, Jena, Germany). The fluorescence intensity of cells with active enzymes of MMP-2 or MMP-9 was quantified by the arithmetic mean intensity of ZEN 2.3 lite software. Each sample was assayed in triplicate, as described previously [29,30,31].

### 4.9. Rat Aortic Ring Assay

Aortas were obtained from six-week-old Sprague–Dawley rats. Each aorta was cut into 1-mm slices and washed in sterile PBS three times, and then imbedded into 55 μL of Matrigel in 96-well plates. The aortic rings were then cultured in 100 μL of DMEM medium with 10% FBS and various concentrations of PSS and PSS with different molecular markers. On day 7, the rings were photographed under a microscope (Colibri 7, ZEISS, Jena, Germany). The data obtained were analyzed and quantified using ImageJ software (ImageJ 1.8.0, Rawak Software Inc., Stuttgart, Germany). The animal experiments were approved by the Animal Ethics Committee of Marine Biomedical Research Institute of Qingdao (MBRI-2017-1106), and were strictly followed the guidelines of the institute.

### 4.10. Chick Chorioallantoic Membrane Assay

Fertilized eggs were incubated in a constant-temperature incubator maintained at 37 °C and 40%–60% humidity for seven days. Gentle suction was applied to the hole located at the broad end of the egg to create a false air sac directly over the chick chorioallantoic membrane (CAM), and a 1–2 cm^2^ segment was immediately removed from the eggshell. A round gelatin sponge (5 mm × 5 mm) saturated with PSS solution (100 or 200 μg/egg) or saline was placed into the area between the pre-existing vessels, and the embryos were further incubated for 48 h. The zones of neovascularization under and around the gelatin sponge were photographed under a stereomicroscope (SZX2-ILLT, OLYMPUS). The data obtained were analyzed and quantified using ImageJ software (ImageJ 1.8.0, Rawak Software Inc., Stuttgart, Germany).

### 4.11. Statistical Analysis

Statistical analysis for in vitro was performed using Excel. A two-tailed Student’s unpaired *t*-test was performed to compare the untreated control group with the treated groups. The results were considered significant when *p* < 0.05 (*, *p* < 0.05; **, *p* < 0.01). Independent experiments were conducted with a minimum of two biological replicates per condition to allow statistical comparison. Error bars represent the standard error of the mean, and the *p* values are indicated. All experiments were repeated at least three times.

## Figures and Tables

**Figure 1 marinedrugs-17-00257-f001:**
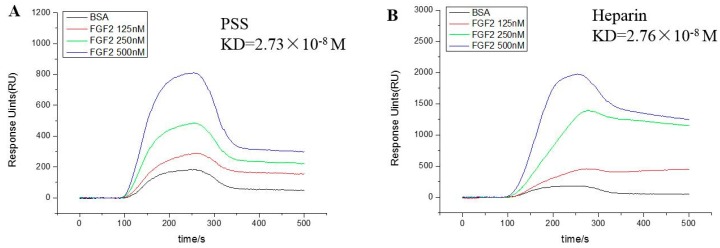
Analysis of the affinity between propylene glycol alginate sodium sulfate (PSS) and fibroblast growth factor 2 (FGF2). (**A**,**B**) were binding response curves of PSS and heparin with FGF2, respectively. PSS or Heparin (1–5 mM) was immobilized on Graft-to-PCL sensor chips. The mobile phase was FGF2 solution (dissolved in phosphate buffer solution (PBS)), and the concentrations were 125, 250, and 500 nM. The data obtained were analyzed and fitted by a PLEXERA SPR Data Analysis Module (DAM) to obtain the equilibrium dissociation constant (KD). Images are representative of three independent experiments with similar results.

**Figure 2 marinedrugs-17-00257-f002:**
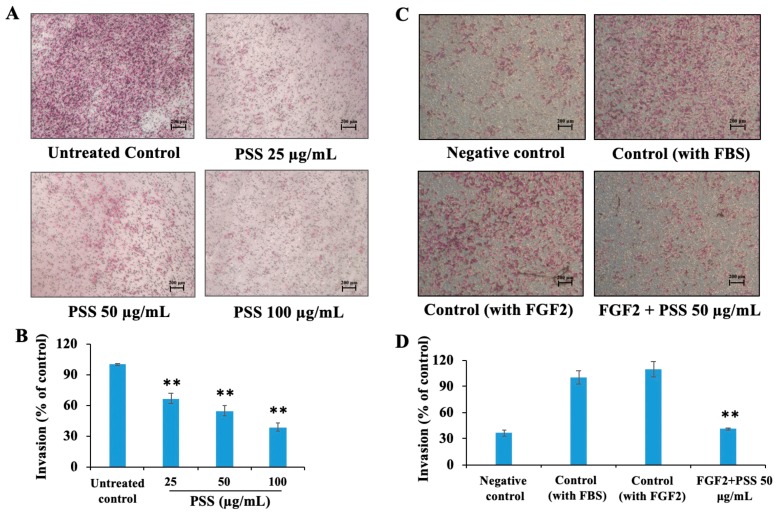
The effect of PSS on the invasion of B16-F10 cells. B16-F10 cells (1.5 × 10^4^ cells/well) were seeded onto a membrane coated with Matrigel (**A**,**B**) or growth factor-reduced Matrigel (**C**,**D**), and were treated with various concentrations of PSS (25, 50, 100 µg/mL) for 16 h or 8 h (FGF2-mediated invasion of B16-F10). Cells that penetrated through to the lower surface of the membrane were stained with crystal violet and photographed under a light microscope at 40× magnification. Then, crystal violet was dissolved in 10% acetic acid, and the absorbance of the resulting solution was measured at 600 nm using a microplate reader (SpetraMAX i3, Molecular Devices, Sunnyvale, CA, USA). The controls included a negative control, complete medium only; control (with 10% FBS) complete medium with 10% FBS; control (with FGF2) complete medium with 200 ng/mL FGF2; and FGF2 + PSS 50 μg/mL complete medium with 200 ng/mL FGF2 and 50 μg/mL PSS. The results are from three independent experiments. ** *p* < 0.01, significant difference between PSS-treated groups and the untreated control by Student’s *t*-test.

**Figure 3 marinedrugs-17-00257-f003:**
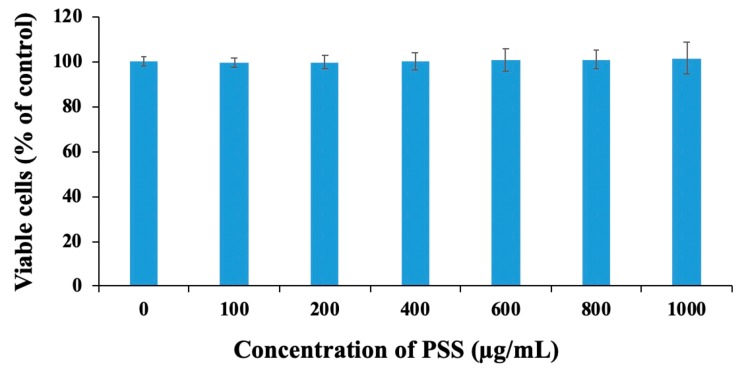
The effect of PSS on cell viability. B16-F10 cells (0.5 × 10^4^ cells/well) were seeded in 96-well plates and incubated for 24 h to allow adherence. Various concentrations of PSS were added to the plates, and then the cells were further incubated for another 48 h. The resazurin assay (1 mg/mL) was used for detection. The absorbance of each well was measured at 405 nm by a microplate reader (SpetraMAX i3, Molecular Devices, Sunnyvale, CA, USA). The data obtained are from three independent experiments.

**Figure 4 marinedrugs-17-00257-f004:**
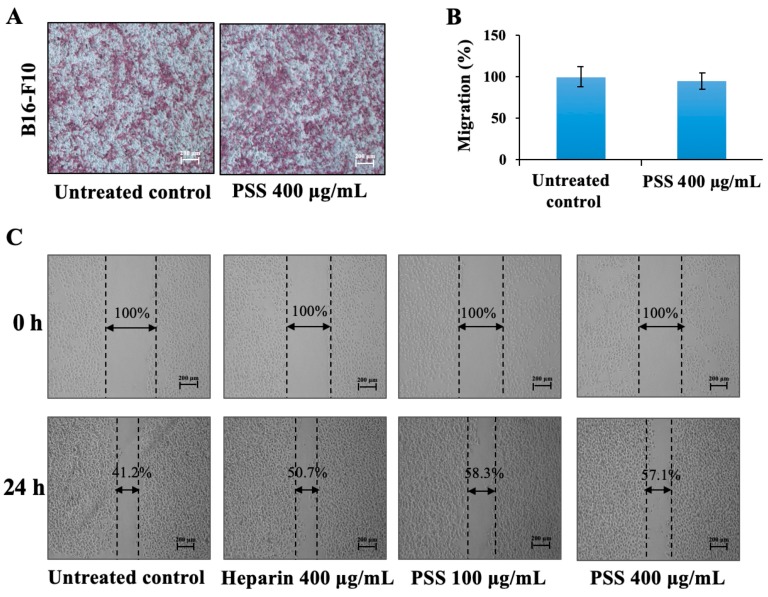
The effect of PSS on the migration of B16-F10 cells. B16-F10 cells (1 × 10^4^ cells/well) were seeded into the upper chamber of Transwell plates, treated with various concentrations of PSS, and allowed to migrate for 16 h. Cells that penetrated through to the lower surface of the membrane were stained with crystal violet and photographed under a light microscope at 40× magnification (**A**). Then, crystal violet was dissolved in 10% acetic acid, and the absorbance of the resulting solution was measured at 600 nm using a microplate reader (SpetraMAX i3, Molecular Devices, Sunnyvale, CA, USA) (**B**). For the wound healing assay (**C**), 70% confluency B16-F10 cells in 12-well culture plates were scratched and then treated with FBS-free medium with PSS (100, 400 μg/mL) or 400 μg/mL heparin. Three randomly selected views along the wound line in each well were photographed under an inverted microscope at 0 h and 24 h after incubation. The percentage of void area with respect to time 0 was determined using ImageJ software (ImageJ 1.8.0, Rawak Software Inc., Stuttgart, Germany). The results are from three independent experiments. ** *p* < 0.01, significant difference between the PSS-treated groups and the untreated control by Student’s *t*-test.

**Figure 5 marinedrugs-17-00257-f005:**
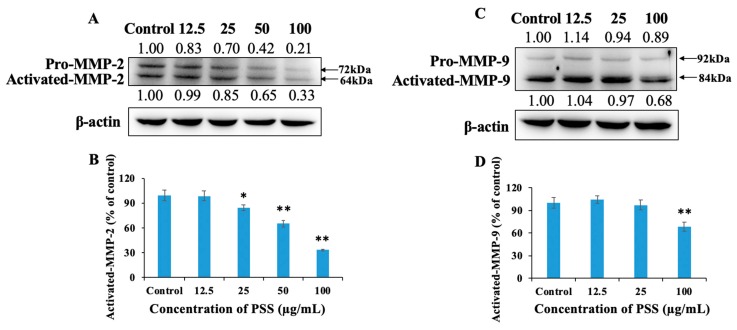
PSS down-regulated the level of activated matrix metalloproteinase 2 (MMP-2) and matrix metalloproteinase 9 (MMP-9) proteins in B16-F10 cells. Cells (2 × 10^6^ cells/dish) were treated with PSS (12.5, 25, 50, 100 µg/mL) for 24 h. Then the cells were harvested, total protein was determined, and SDS-PAGE was performed, as described in the Materials and Methods section. The levels of activated MMP-2 (**A**,**B**) and MMP-9 (**C**,**D**) were estimated by Western blotting, also as described in the Materials and Methods section. The numbers underneath the blots represent the band intensities (normalized to the loading controls, means of three independent experiments) measured by ImageJ software (ImageJ 1.8.0, Rawak Software Inc., Stuttgart, Germany). The standard deviations were all within ±15% of the means (data not shown); β-actin was used as an equal loading control. The experiments were repeated three times.

**Figure 6 marinedrugs-17-00257-f006:**
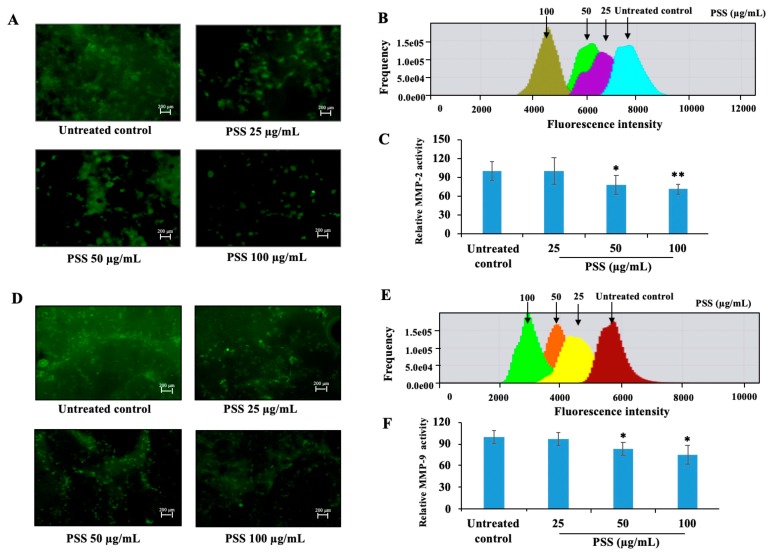
The effect of PSS on the activity of MMP-2 and MMP-9 in B16-F10 cells. The activity of MMP-2 (**A**–**C**) and MMP-9 (**D**–**F**) was tested using the GENMED Kit (Genmed Scientifics Inc., Wilmington, DE, USA). Cells (1 × 10^5^ cells/well) were seeded in 12-well culture plates with a glass slide to reach 50% confluence. After being washed with PBS, fresh serum-free culture medium was added to the plate in the presence or absence of serial treatments of PSS (25, 50, 100 μg/mL) for 24 h. The GENMED kit was used to detect the fluorescence intensity. The green dots represent the catalytic sites for MMP-2 or MMP-9, and the fluorescence intensity represents the amount of the degraded gelatin. Fluorescence was visualized in 10 randomly selected fields of view for each cell slide under a fluorescence microscope (Colibri 7, ZEISS, Jena, Germany), and the fluorescence intensity was quantified by ZEN 2.3 lite software (ZEISS, Jena, Germany). The data shown above are from three independent experiments. * *p* < 0.05, ** *p* < 0.01, significant difference between the PSS-treated groups and the untreated control by Student’s *t*-test.

**Figure 7 marinedrugs-17-00257-f007:**
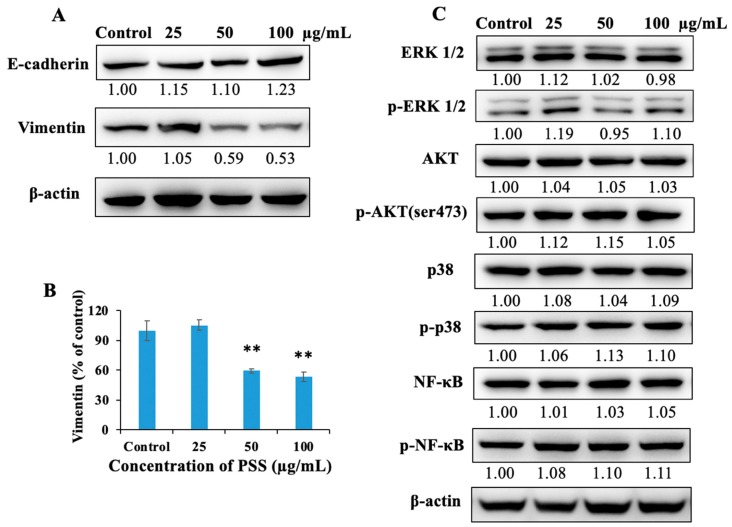
PSS down-regulated the protein expression of Vimentin in B16-F10 cells. Cells (2 × 10^6^ cells/dish) were treated with PSS (25, 50, 100 µg/mL) for 24 h. The cells were harvested, total protein was determined, and SDS-PAGE was performed as described in the Materials and Methods section. The levels of ERK 1/2, p-ERK 1/2, AKT, p-AKT, p38, p-p38, NF-κB, p-NF-κB, Vimentin, and E-cadherin were estimated by Western blotting, as described in the Materials and Methods section. (**A**,**B**) The protein expression level of Vimentin and E-cadherin in B16-F10 cells. (**C**) The levels of other proteins mentioned above in B16-F10 cells. The numbers underneath the blots represent the band intensities (normalized to the loading controls, means of three independent experiments) measured by ImageJ software (ImageJ 1.8.0, Rawak Software Inc., Stuttgart, Germany). The standard deviations were all within ±15% of the means (data not shown); β-actin was used as an equal loading control. The experiments were repeated three times.

**Figure 8 marinedrugs-17-00257-f008:**
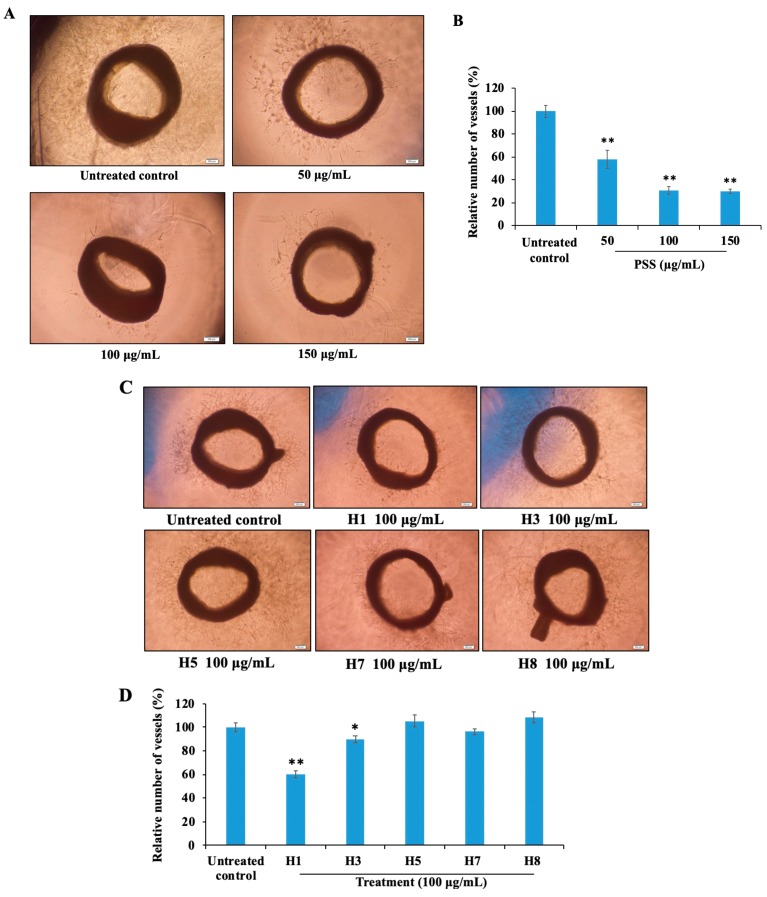
PSS suppresses angiogenesis. The inhibitory effect of PSS on microvessel outgrowth arising from rat aortic rings. Aortic rings were embedded in Matrigel in 96-well plates and then cultured with medium containing various concentrations of PSS (50, 100, 150 μg/mL) for seven days (**A**,**B**). The effect of fractionated PSS on angiogenesis in the rat aortic ring model. The method used was the same as above (**C**,**D**). The effect of PSS on angiogenesis in the chick chorioallantoic membrane assay. Gelatin sponges (5 mm × 5 mm) saturated with various concentrations of PSS solution or normal saline were inserted into seven-day-old fertilized eggs (**E**,**F**). After incubation for another 48 h, the zones of neovascularization under and around the gelatin sponge were photographed under an anatomic microscope (Colibri 7, ZEISS, Jena, Germany) and the black arrows in the photos pointed at the site of gelatin sponge. The total vessel number was quantified by ImageJ software (ImageJ 1.8.0, Rawak Software Inc., Stuttgart, Germany). * *p* < 0.05, ** *p* < 0.01, significant difference between the PSS or PSS fraction-treated groups and the untreated control by the Student’s *t*-test.

**Figure 9 marinedrugs-17-00257-f009:**
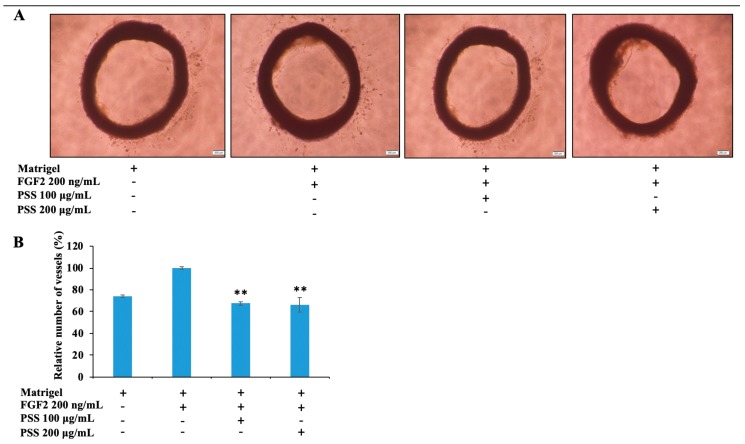
PSS suppresses FGF2-induced microvessel outgrowth in a rat aortic ring model. (**A**) Aortic rings were embedded in growth factor-reduced Matrigel in 96-well plates, and were then cultured with medium containing 200 ng/mL FGF2 and various concentrations of PSS (100, 200 μg/mL) for seven days. Images are representative of three independent experiments with similar results. (**B**) The total vessel number was quantified by ImageJ software (ImageJ 1.8.0, Rawak Software Inc., Stuttgart, Germany). ** *p* < 0.01, significant difference between the PSS-treated groups and the FGF2 control by Student’s *t*-test.

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
