# Peer review of "The Inhibitory Effect of Propylene Glycol Alginate Sodium Sulfate on Fibroblast Growth Factor 2-Mediated Angiogenesis and Invasion in Murine Melanoma B16-F10 Cells In Vitro"

_marinedrugs, 2019, doi:10.3390/md17050257_

Round 1

Reviewer 1 Report

General comment :

This is an interesting in vitro study showing that propylene glycol alginate sodium sulfate can reduce “invasion” of murine melanoma B16-F10 Cells cells and angiogenesis. Some robust and quantitative in vitro experiments are lacking. In vivo experiment should be performed and presented along with these results.

Specific comments :

Abstract : please define B16-F10 cells

Text: The B16-F10 definition appears very late in the manuscript. It should be present in the introduction. In general, all kind of cells used in this study should be introduced in the introduction.  

Line 95: “PSS inhibited invasion by 33.81%, 45.7% and 61.8%”. Which assay was used to detect “invasion”? This should be mentioned in the results. “33.81%”, 33.8% is probably sufficiently precise, please correct throughout the manuscript.

Figure 2: it would be interesting to see if PSS induced decreased proliferation using BrDU or EDU staining.

Figure 5: it would be interesting to assess this effect on MMP9 and MMP2 in another type of cell to demonstrate the specificity (or not) of this finding.

Figure 6: can the GENMED Kit distinguish between overall decrease activity of MMPs due to the decreased number of cells and true decrease in activity? A qPCR would be easy to do here and would give a much more precise answer.

Figure 8C: this panel is not completely convincing. Differences are hard to see.

Figure 8: authors should provide quantifications

Figure 9: authors should provide quantifications. Can the authors provide a control, non-treated with FGF, but treated with PSS to make sure the effect is specific to FGF?

In vivo studies with tumor implanted in under the skin or in the liver of mice and +/- treated with PSS would be interesting.

The link between angiogenesis and the effect on B16-F10 cells is lacking or is not clearly explained/investigated.

Author Response

Point-by-point response

Addressing both editor and reviewers’ concern has greatly enhanced the quality of our manuscript. We thank you for your contributions to our work.

Reviewers' comments:

Reviewer #1

General comment:

This is an interesting in vitro study showing that propylene glycol alginate sodium sulfate can reduce “invasion” of murine melanoma B16-F10 Cells cells and angiogenesis. Some robust and quantitative in vitro experiments are lacking. In vivoexperiment should be performed and presented along with these results.

Specific comments:

(1)  Please define B16-F10 cells

Response:We have defined B16-F10 cells as “murine melanoma B16-F10 cells” when first used.

(2)  Text: The B16-F10 definition appears very late in the manuscript. It should be present in the introduction. In general, all kind of cells used in this study should be introduced in the introduction.  

Response: We have introduced B16-F10 cells in the introduction section in line 84.

(3)  “PSS inhibited invasion by 33.81%, 45.7% and 61.8%”. Which assay was used to detect “invasion”? This should be mentioned in the results. “33.81%”, 33.8% is probably sufficiently precise, please correct throughout the manuscript.

Response:The invasion was detected by the classical Matrigel-coated transwell assay. We have changed “33.81%” to “33.8%” and kept all the relative results to same precision though the manuscript.

(4)  Figure 2: it would be interesting to see if PSS induced decreased proliferation using BrDU orEDU staining.

Response: We have quantified the effect of PSS on the proliferation of B16-F10 cells by resazurin assayand no obvious decrease on the growth was detected (As shown in Figure 3). The resazurin assay a reliable and robust method, which was frequently used to measure cell proliferation and viability in high-throughput screening (HTS) attempts. As BrDU, EDU staining and resazurin assay were all methods to detect proliferation, they should obtain the same trend with the same treatment of PSS. 

(5)  Figure 5: it would be interesting to assess this effect on MMP9 and MMP2 in another type of cell to demonstrate the specificity (or not) of this finding.

Response: We measured the effect of PSS on migration of B16-F10, A549 and HCT116 cells and no obviously inhibitory effect was observed. We further detected the effect of PSS on invasion of B16-F10 and A549 and the results showed that PSS dose-dependently decreased the invasion of B16-F10 cells while no inhibitory effect was found in A549 cells. Invasion of tumor cells was biological process that was to firstly degrade the cellular matrix by MMPs and then to migrate somewhere. As PSS inhibited the invasion of B16-F10 cells but not A549 cells, so we speculated PSS might affect the levels or activities of MMP-2 or MMP-9 in B16-F10 cells but no in A549 cells. 

(6)  Figure 6: can the GENMED Kit distinguish between overall decrease activity of MMPs due to the decreased number of cells and true decrease in activity? A qPCR would be easy to do here and would give a much more precise answer.

Response: TheGENMED Kit was commercially developed to specially detect the activated MMP-2 and MMP-9 in situ. The green dots in the images represented the catalytic sites for MMP-2 or MMP-9 and the fluorescence intensity represented the amount of the degraded gelatin. The higher florescence intensity reflected the the stronger activities of MMP-2 or MMP-9. In the present study, the B16-F10 cells was treated by serial concentration of PSS for 24 h, and then was handled according to the standard protocol of the GENMED Kit to detect the relative change of the fluorescence intensity after the treatment of PSS. As there was no detectable inhibition of PSS on the number of cancer cells (As shown in Figure 3), so the decrease of the fluorescence intensity was supposed to reflect the true decrease of activity of MMP-2 or MMP-9. 

(7)  Figure 8C: this panel is not completely convincing. Differences are hard to see. 

Response: We have quantified all the panels using the software of Image J. The relative inhibition was shown in Figure 8C and 8D.

(8)  Figure 8: authors should provide quantifications

Response:We have quantified all the panels using the software of Image J. The relative inhibition was shown in Figure 8B, 8D and 8F respectively.  

(9)  Figure 9: authors should provide quantifications. Can the authors provide a control, non-treated with FGF, but treated with PSS to make sure the effect is specific to FGF? 

Response:We have quantified all the panels using the software of Image J. The relative inhibition was shown in Figure 9A and 9B. The “Control” group in Figure 9 was the negative control that was untreated with FGF2. We have changed the legends of control groups in Figure 9 as Control (non-treated with FGF2) and Control (treated with FGF2).

(10) In vivo studies with tumor implanted in under the skin or in the liver of mice and +/- treated with PSS would be interesting.

Response: Based on the reports internationally and our previous research, sulfated polysaccharides including PSS have been proven to exert significant advantages in improving the tumor environments to prolong survival for cancer patients by reducing blood viscosity, blocking the adhesion, inhibiting angiogenesis, suppressing the activities of MMPs, inhibiting the invasion and so on. Unlike the chemotherapeutic drugs, sulfated polysaccharides including PSS couldn’t directly kill cancer cellsin vitro and induce obvious decrease of the volume of tumor in vivo. So, it would be an attractive anti-tumor strategy to combine sulfated polysaccharides with chemotherapeutic drugsto synergistically enhance therapeutic efficacy for chemotherapy.

(11)The link between angiogenesis and the effect on B16-F10 cells is lacking or is not clearly explained/investigated.

Response: Tumor microenvironment refers to the close relationship between the occurrence, growth and metastasis of tumor and the internal and external environment of tumor cells, including the proliferation of tumor cells, the migration and invasion of tumor cells, angiogenesis, the adhesion, EMT and so on. B16-F10 cells was one highly metastatic models and overexpressed FGF2 growth factor. Our data indicated that PSS suppressed FGF2-mediated invasion and angiogenesis.

Reviewer 2 Report

He Ma et all have investigated the in vitro effect of Propylene Glycol Alginate Sodium Sulfate (PSS) on the different features of metastasis – cell invasion, cell migration and angiogenesis. The PSS impact on the proteins involved in the regulation of invasion and migration as well as tumor microenvironment e.g. FGF2, MMP-2/9, Vimentin was also assessed. The overall findings are interesting and deliver new data on the PSS biological activity. The data on anti-metastatic activity of PSS are solid and include several experiments covering different processes crucial for metastasis.  However, it is a pity that the article is missing the study showing the impact of PSS on the tumor development and metastasis in animal model. The mechanism of action could be proposed or at least discussed. There is a lack of conclusion. Overall impression is that the article is careless and cursory.

My remarks:

The manuscript needs to improve language.

The discussion should be improved.

The conclusions are too far-reaching: e.g page 2 line 82: These data indicated that FGF2 was a potential target in determining the effect of PSS on angiogenesis. This conclusion needs at least explanation.

Abbreviation should be explained when first used.

Although the origin of PSS is known for the authors it would be beneficial when the authors explain PSS connection with marine habitats in the introduction section

When authors mention B16-F10 cells it should be at least one time explained what kind of cells they are. Why the authors decided to study murine cell line and compared the results with the human cancer cell lines?

Why A549 cells were studied – are they FGF2 – negative? It should be explained.

Results:

General remarks: The results part needs to be more explanatory, e.g. why these particular tests were done, why authors decided to study these particular proteins? Please, add to Figure Captions the information that  mean and SEM are presented in the graphs. Figure captions: (a) and (b) references to the graphs are missing.

Figure 2. What is the difference between control and negative control?

Figure 3. Why different cells were used in each experiment; how can you compare and discuss result obtained in different cell lines? Why MTT test was used while in Fig 1 the crystal violet staining. These tests could deliver different results (Śliwka et al PlosOne 2016). MTT test is described not properly in the materials and methods sections. Otherwise, the method of test conducting was wrong and the results are incorrect.

Line 80. Explain what VEGF165, where are corresponding results, please explain why in the Figure 1 caption VEGF165 is mentioned?

Line 131.What is it ECM – please provide the full name

Paragraph 2.4. The results description is unintelligible

Figure 4. What kind of cells are HCT116? Why they were introduced into the study?

Paragraph 2.5. Why MMP-2 and MMP-9 - these two proteins were studied?  The following sentences are not logical: “Based on the data above, we confirmed that PSS inhibited the invasion of B16-F10 cells in a dose-dependent manner but had no effect on tumor cell migration. Therefore, we examined whether PSS affected the expression or activity of crucial ECM-degrading enzymes.” It was shown before that PSS had no effect of B16-F10 cells migration, either (Fig 4 B, Supplementary data).

Figure 6. The microscopic images are of bad quality. Please explain what represents green dots in the image. The scale bar is missing. What represents x and y axis in the histograms.

Paragraph 2.8. The authors state that they studied in this experiment PSS fractions. Why? Why only in this experiment?

Figure 8. Authors placed the black arrow in the image. What is it pointing at? The scale bars are missing.

Figure 9. The scale bar is missing.

Discussion

General remark: a majority of discussion concerns a binding constant and possible mechanism of PSS - FGF2 interaction. There is a lack of such detailed discussion on the remaining results which are a majority of the study.

Line 236: “In particular, we studied the activity of PSS on the effects triggered by FGF2.” Please explain how your experiments prove FGF2-triggered effect (except of 2.1, 2.2, 2.9)

Line 239-240 “Moreover, PSS could suppress the FGF2-mediated invasion but not the migration of B16-F10 cells.” Please support this statement with the appropriate results.

Line 248-249. FGF2 bind to PSS or PSS bind to FGF2?

Line 250-251. The Figure 10 and its description should be moved to Results section. Otherwise, it should be stated that these results are taken from literature.

Lines 276-281. This part fits better to Introduction; moreover the results in FGHF2-negative line should be performed to support the statement.

Line 282-289. This paragraph is unnecessary

Lines 291-294. “We found that PSS itself could not inhibit the proliferation of tumor cells, and its effect on tumor cells was mainly dependent on the suppression of FGF2-mediated angiogenesis and invasion as well as blockade of the binding of tumor cells with adhesive molecules [16].” In my opinion this conclusion is not sufficiently justified.

Materials and methods

General remark: The Material and Methods section should be ordered according to results section

4.2. Why the fluorescence was measured in spectrophotometer? Why the resazurin solution was used in the test, in corresponding figure captions authors state that MTT test was applied?

4.9. Please explain the clue of this test

4.11. Student t-test is used for two groups comparison, for more groups ANOVA test with post-hoc test is appropriate.

Author Response

Point-by-point response

Addressing both editor and reviewers’ concern has greatly enhanced the quality of our manuscript. We thank you for your contributions to our work.

Reviewers' comments:

Reviewer #2

General comment:

He Ma et all have investigated the in vitro effect of Propylene Glycol Alginate Sodium Sulfate (PSS) on the different features of metastasis – cell invasion, cell migration and angiogenesis. The PSS impact on the proteins involved in the regulation of invasion and migration as well as tumor microenvironment e.g. FGF2, MMP-2/9, Vimentin was also assessed. The overall findings are interesting and deliver new data on the PSS biological activity. The data on anti-metastatic activity of PSS are solid and include several experiments covering different processes crucial for metastasis.  However, it is a pity that the article is missing the study showing the impact of PSS on the tumor development and metastasis in animal model. The mechanism of action could be proposed or at least discussed. There is a lack of conclusion. Overall impression is that the article is careless and cursory. 

My remarks:

(1)  The manuscript needs to improve language. 

Response: The manuscript was further polished by a commercial editing company in USA.

(2)  The discussion should be improved.

Response:We have rewritten some of the Discussion section to improve the logicality.

(3)  The conclusions are too far-reaching: e.g page 2 line 82: These data indicated that FGF2 was a potential target in determining the effect of PSS on angiogenesis. This conclusion needs at least explanation.

Response:We have changed the sentence to “As FGF2 is crucial growth factor to regulate angiogenesis and the function of tumor cells, so, these data indicated that FGF2 was probably a potential target for PSS to improve tumor environment.” in line 93-95.

(4)  Abbreviation should be explained when first used.

Response: We have modified B16-F10 cells and ECM to murine melanoma B16-F10 cells and extracellular matrix (ECM) respectively when first used.

(5)  Although the origin of PSS is known for the authors it would be beneficial when the authors explain PSS connection with marine habitats in the introduction section

Response: We have added the sentence “PSS is obtained from alginate polysaccharide of Laminaria with multiple chemical modifications.” to explain the connection of PSS with marine habitats in the introduction section in line 74-75.

(6)  When authors mention B16-F10 cells it should be at least one time explained what kind of cells they are. Why the authors decided to study murine cell line and compared the results with the human cancer cell lines?

Response:We have modified B16-F10 cells to “murine melanoma B16-F10 cells” when first used. We have focused on the effect of PSS on highly metastatic B16-F10 melanoma model and deleted the data about other cancer cell lines including HCT116, A549 and LS180. 

(7)   Why A549 cells were studied – are they FGF2 – negative? It should be explained. 

Response: We have deleted the data about A549 cell line.

 Results:

General remarks: The results part needs to be more explanatory, e.g. why these particular tests were done, why authors decided to study these particular proteins? Please, add to Figure Captions the information that mean and SEM are presented in the graphs. Figure captions: (a) and (b) references to the graphs are missing. 

ResponseThank you for your good suggestions. We have carefully checked all of the results parts and the figure captions and have made necessary modifications. 

(8)  Figure 2. What is the difference between control and negative control?

ResponseThe “control” in Figure 2A was the complete medium containing 10% FBS. The “negative control” was just the complete medium without FBS. To make easier to understand, we have renamed the “control” in Figure 2A to “Untreated control”. Meanwhile, as shown in Figure 2B, we have renamed the legends of “Negative control, 10% FBS, FGF2 200 ng/mL” to “Negative control, Control (with 10% FBS), Control (with FGF2)” respectively. We also made detailed explanations about the difference in the legend of Figure 2. Additionally, we changed all the “Control” to “Untreated control” in the Figures and throughout the manuscript.

(9)   Figure 3. Why different cells were used in each experiment; how can you compare and discuss result obtained in different cell lines? Why MTT test was used while in Fig. 1 the crystal violet staining. These tests could deliver different results (Śliwka et al PlosOne 2016). MTT test is described not properly in the materials and methods sections. Otherwise, the method of test conducting was wrong and the results are incorrect.

Response:Thank you for your careful reading. We made a mistake about methodical descriptions in this part. The method we used to measure the viability of cancer cells after the treatment of PSS was the resazurin assay but not the MTT assay. The resazurin assay a reliable and robust method, which was frequently used to measure cell proliferation and viability in high-throughput screening (HTS) attempts. We have changed the “multiwell spectrophotometer” to “microplate reader” and revised “The MTT assay (5 mg/mL)” to “Theresazurin assay (1 mg/mL)” in legend of Figure 3. 

(10)Line 80. Explain what VEGF165, where are corresponding results, please explain why in the Figure 1 caption VEGF165 is mentioned?

Response:Thank you for your careful reading. We made a mistake in the caption of Figure 1. We have deleted VEGF in the caption of Figure1. Both FGF2 and VEGF165 are crucial growth factors associated with angiogenesis during cancer Pprogress, so, we detected the binding affinity of PSS with VEGF165 and FGF2. Based on the data obtained from SPR assay, PSS exhibited higher affinity to FGF2 (KD=2.73×10-8 M) than that of VEGF165 (KD=1.78×10-4M), suggesting that FGF2 might be the potential target for PSS. Binding response curves of interactions between PSS with VEGF165 was shown in Supplementary Table 1.

(11) Line 131.What is it ECM – please provide the full name

Response: We have added the full name of ECM to extracellular matrix(ECM) when first used.

(12)Paragraph 2.4. The results description is unintelligible

Response: We have rearranged the figures related to migration and described the corresponding results in the Paragraph 2.4 in line 227-232.

(13)Figure 4. What kind of cells are HCT116? Why they were introduced into the study? 

Response:HCT116 cells is from human colon cancer patients. We have deleted the data about HCT116 cell line.

(14)Paragraph 2.5. Why MMP-2 and MMP-9 - these two proteins were studied?  The following sentences are not logical: “Based on the data above, we confirmed that PSS inhibited the invasion of B16-F10 cells in a dose-dependent manner but had no effect on tumor cell migration. Therefore, we examined whether PSS affected the expression or activity of crucial ECM-degrading enzymes.” It was shown before that PSS had no effect of B16-F10 cells migration, either (Fig. 4). 

Response:Degradation of the basement membrane is an essential step for the metastatic progression of most cancers. Type IV collagen was the most abundant component of the basement membrane. Matrix metalloproteinase-2 (MMP-2) and Matrix metalloproteinase-9 (MMP-9) are two major degrading enzymes to type IV collagen. So, MMP-2 and MMP-9 was potential targets to treat cancer. The matrigel used in the invasion process was to design to mimic the components of basement membrane. PSS was shown to block the degradation of matrigel by B16-F10 cells. Therefore, it is necessary to detect whether PSS affect the expressions and activities of MMP-2 and MMP-9 proteins. 

(15)Figure 6. The microscopic images are of bad quality. Please explain what represents green dots in the image. The scale bar is missing. What represents x and y axis in the histograms. 

Response: We have added the scale bars in all microscopic images. The x axis represents the fluorescence intensity, the y axis represents frequency as well as cell density. The green dots represented the catalytic sites for MMP-2 or MMP-9 and the fluorescence intensity represented the amount of the degraded galetin. The higher florescence intensity reflected the the stronger activities of MMP-2 or MMP-9. We also have added the explanation to the legend of Figure 6.

(16)Paragraph 2.8. The authors state that they studied in this experiment PSS fractions. Why? Why only in this experiment?

ResponseIt was well-documented that the molecular weight of sulfated polysaccharides including PSS played crucial roles in determining their bioactivities. Our previous study showed that the average molecular weight of PSS was about 17 kDa and the distribution range of molecular weight was about 2 - 20 kDa. Unlike the small molecular compounds, which generally bind to domains with catalytic activity of targeting proteins, sulfated polysaccharides possessed large amount of negative charge and generally interacted with proteins rich in positive potential on the surface of proteins. Theoretically, the longer sugar chains (the higher molecular weight) of the negative charged polysaccharide was endowed with the stronger binding affinity to target proteins and further exerted obvious bioactivities. In this project, the higher molecular weight exerted stronger inhibitory effect on angiogenesis. It might be the same trends of PSS fractions in other experiments of the manuscript. 

(17)Figure 8. Authors placed the black arrowin the image. What is it pointing at? The scale bars are missing.

ResponseWe have added the scale bars in all microscopic images. The black arrow was pointing at gelatin sponge, which was the carrier of the treatment.

(18)Figure 9. The scale bar is missing.

Response:We have added the scale bars in all microscopic images.

Discussion

General remark: a majority of discussion concerns a binding constant and possible mechanism of PSS - FGF2 interaction. There is a lack of such detailed discussion on the remaining results which are a majority of the study.

(19)Line 236: “In particular, we studied the activity of PSS on the effects triggered by FGF2.” Please explain how your experiments prove FGF2-triggered effect (except of 2.1, 2.2, 2.9)

Response: We have re-edited the sentence in the first paragraph of the Discussion section as “In this study, we observed that PSS has a major impact on invasion and angiogenesis in murine melanoma B16-F10 cells. FGF2 is a proangiogenic factor involved in tumor angiogenesis, invasion and migration.”

(20)Line 239-240 “Moreover, PSS could suppress the FGF2-mediated invasion but not the migration of B16-F10 cells.” Please support this statement with the appropriate results.

Response: We have changed the sentence to “Moreover, PSS could suppress the FGF2-mediated invasion” in line 483-484.

(21)Line 248-249. FGF2 bind to PSS or PSS bind to FGF2?

Response: The high affinity between FGF2 and PSS was probably due to the interaction of positive charges on the surface of FGF2 and the negative charges on the surface of PSS. Thus, FGF2 could bind to PSS and PSS could also bind to FGF2. FGF2 bound to PSS” will be better. in line 24, 91, 488.

(22)Line 250-251. The Figure 10 and its description should be moved to Results section. Otherwise, it should be stated that these results are taken from literature. 

Response: We have moved the Figure 10 to supplementary data as Supplementary Figure there. The figure showed the distribution of the positive potential on the surface of FGF2 and VEGF165. As shown in the figure, the positive potential on the surface of FGF2 was tend to be concentrated while the positive potential on the surface of VEGF165 was dispersed. The binding of PSS with FGF2 or VEGF165 was probably due to the interaction of positive and negative charges, and concentrated positive charges was likely to indicate more firm binding. Thus, the figure might help to illustrate the distinguished affinity between FGF2 and VEGF165 to PSS.

(23)Lines 276-281. This part fits better to Introduction; moreover the results in FGF2-negative line should be performed to support the statement. Line 282-289. This paragraph is unnecessary()

Response: The content of Lines 276-281 mainly discussed the distinguishing expression of FGF2 between B16-F10 and A549 cells. To mainly focus on the effect of PSS on B16-F10 cells, we have deleted the data about other cancer cell line, including A549 cells. So we have deleted Lines 276-281 from the Discussionsection. Moreover, we also deleted Line 282-289 from the Discussionsection. Thank you for your suggestion to set the negative cell line for scientific research. It is an important golden rule to improve the logicality of the study, which will help me to design my future project with high logicality and more rigorous.

(24)Lines 291-294. “We found that PSS itself could not inhibit the proliferation of tumor cells, and its effect on tumor cells was mainly dependent on the suppression of FGF2-mediated angiogenesis and invasion as well as blockade of the binding of tumor cells with adhesive molecules [16].” In my opinion this conclusion is not sufficiently justified.

Response: We have changed the last paragraph in Discussionsection to “For the first time, we evaluated the effect of PSS on the highly metastatic B16-F10 melanoma cells and the related tumor environment. PSS itself have no inhibitory effect on the growth of B16-F10 cells, however, it suppressed FGF2-mediated angiogenesis and invasion of B16-F10 cells as well as decreased the level of Vimentin which might help to enhance the sensitivity of tumor cells to chemotherapy. Moreover, to fully elucidate the effects of PSS on the tumor microenvironment, further research should be conducted to investigate whether PSS exerts inhibitory effects on other cells involved in the tumor microenvironment, such as endothelial cells, fibroblasts, and immunecells. Meanwhile, further research should be done to combine PSS with chemotherapeutic drugs to check whether synergistical effect happens.”

Materials and methods

(25)General remark: The Material and Methods section should be ordered according to results section

Response:We have revised the order of the “Material and Methods” to match the order of the “Results” sections.

(26)4.2. Why the fluorescence was measured in spectrophotometer? Why the resazurin solution was used in the test, in corresponding figure captions authors state that MTT test was applied? 

Response: Thank you for your careful reading. We made a mistake about methodical descriptions in this part. We have changed the“multiwell spectrophotometer” to “microplate reader”. The method we used to measure the viability of cancer cells after the treatment of PSS was the resazurin assay but not the MTT assay. The resazurin assay a reliable and robust method, which was frequently used to measure cell proliferation and viability in high-throughput screening (HTS) attempts.

(27)4.9. Please explain the clue of this test

Response:TheGENMED Kit was commercially developed to specially detect the activated MMP-2 and MMP-9 in situ. Gelatin labeled with fluorescein isothiocyanate (FITC) was used as a substrate and a specific inhibitor of MMP-2 or MMP-9 was alternatively added when needed. The green dots in the images represented the catalytic sites for MMP-2 or MMP-9 and the fluorescence intensity represented the amount of the degraded gelatin. The higher florescence intensity reflected the the stronger activities of MMP-2 or MMP-9. In the present study, the B16-F10 cells was treated by serial concentration of PSS for 24 h, and then was handled according to the standard protocol of the GENMED Kit to detect the relative change of the fluorescence intensity after the treatment of PSS. As there was no detectable inhibition of PSS on the number of cancer cells (As shown in Figure 3), so the decrease of the fluorescence intensity was supposed to reflect the true decrease of activity of MMP-2 or MMP-9. 

(28)Student t-test is used for two groups comparison, for more groups ANOVA test with post-hoc test is appropriate.

Response:In the study, a two-tailed Student’s unpaired t-test was used to compare the differences between untreated group and the treated groups. As we didn’t compare the differences among more groups simultaneously, so we didn’t run the ANOVA test with post-hoc test.

Reviewer 3 Report

 The current study provides some interesting observations. It is technically sound but there are certain major concerns that need to be addressed.

·       What was the rationale of using mouse melanoma cells when human melanoma cells are available?

·       Provide scale bars for images shown in Figure 2, 4 and 6.

·       Effect of PSS should be evaluated on levels of activity of MMP-2 and MMP-9 using gelatin zymography.

·       Provide a new blot image for E-Cadherin shown in Figure 7A. Also provide densitometric analysis for panel C.

·       Provide a quantification of data from CAM assay.

·       It will be interesting to see the in vivo effect of PSS (experimental metastasis or subcutaneous injection). Further, corroborating the above findings using markers for invasion, EMT and angiogenesis (e.g. PECAM) by performing the immuno-histochemistry.

·       The references should be updated to include latest global trends for melanoma.

Author Response

Point-by-point response

Addressing both editor and reviewers’ concern has greatly enhanced the quality of our manuscript. We thank you for your contributions to our work.

Reviewer #3

 The current study provides some interesting observations. It is technically sound but there are certain major concerns that need to be addressed.

(1)  What was the rationale of using mouse melanoma cells when human melanoma cells are available?

Response: B16-F10 cells is highly metastatic melanoma model and have been widely used to evaluate the anti-tumor potency for compounds in vitro. Meanwhile, more cancer related models in vivohave been developed using B16-F10 cell line to match the process in clinic.

(2)  Provide scale bars for images shown in Figure 2, 4 and 6.

Response: We have added the scale bars in all microscopic images.

(3)  Effect of PSS should be evaluated on levels of activity of MMP-2 and MMP-9 using gelatin zymography.

Response:TheGENMED Kit was commercially developed to specially detect the activated MMP-2 and MMP-9 in situ. The bands obtained from the gelatin zymography assay might include the activated MMP-2 or MMP-9 and pro-MMP-2 or MMP-9 due to the incomplete activation. The results obtained from the GENMED Kit could also factually reflected the enzymatic activities of MMP-2 and MMP-9 like the data from the gelatin zymography assay.

(4)  Provide a new blot image for E-Cadherin shown in Figure 7A. Also provide densitometric analysis for panel C.

Response: We have repeated to run the western blot experiment for E-cadherin protein and updated the new band in Figure 7A. Meanwhile, we have provided densitometric analysis for the bands in Figure 7C.

(5)  Provide a quantification of data from CAM assay.

Response:The relative inhibition of PSS on CAM assay was quantitatively analyzed using image J software. 

(6)  It will be interesting to see the in vivo effect of PSS (experimental metastasis or subcutaneous injection). Further, corroborating the above findings using markers for invasion, EMT and angiogenesis (e.g. PECAM) by performing the immuno-histochemistry.

Response: Based on the reports internationally and our previous research, sulfated polysaccharides including PSS have been proven to exert significant advantages in improving the tumor environments to prolong survival for cancer patients by reducing blood viscosity, blocking the adhesion, inhibiting angiogenesis, suppressing the activities of MMPs, inhibiting the invasion and so on. Unlike the chemotherapeutic drugs, sulfated polysaccharides including PSS couldn’t directly kill cancer cellsin vitro and induce obvious decrease of the volume of tumor in vivo. So, it would be an attractive anti-tumor strategy to combine sulfated polysaccharides with chemotherapeutic drugsto synergistically enhance therapeutic efficacy for chemotherapy.

(7)  The references should be updated to include latest global trends for melanoma. 

Response: We have updated the latest reference, which was focused on FGF2 related therapy for melanoma disease.

Round 2

Reviewer 1 Report

The authors have adequately responded to my comments

Reviewer 2 Report

The manuscript was improved, however not sufficiently. The major concerns are:

- Discussion should be enhanced with respect to the results apart from the binding assay ( a majority of discussion concerns a binding constant and possible mechanism of PSS - FGF2 interaction. There is a lack of such detailed discussion on the remaining results which are a majority of the study and this should be delivered.)

-Figure captions: (a) and (b) references to the graphs are still missing e.g. in Fig 9 but there are more. The authors did not explained in the text what black arrow point at in Fig 8.

-  The following question was not explained in the text: (16)Paragraph 2.8. The authors state that they studied in this experiment PSS fractions. Why? Why only in this experiment?

Authors state that " In this project, the higher  molecular weight exerted stronger inhibitory effect on angiogenesis. It  might be the same trends of PSS fractions in other experiments of the  manuscript." why such experiments were not performed. At least it should be discussed in the article.

Reviewer 3 Report

The authors have provide justifications for the concerns raised. Overall the manuscript is in good shape.